# Light-fuelled freestyle self-oscillators

Hao Zeng [1]*, Markus Lahikainen[1], Li Liu[2], Zafar Ahmed[1], Owies M. Wani[1], Meng Wang [2], Hong Yang [2] & Arri Priimagi [1]*

Self-oscillation is a phenomenon where an object sustains periodic motion upon non-periodic stimulus. It occurs commonly in nature, a few examples being heartbeat, sea waves and fluttering of leaves. Stimuli-responsive materials allow creating synthetic self-oscillators fuelled by different forms of energy, e.g. heat, light and chemicals, showing great potential for applications in power generation, autonomous mass transport, and self-propelled micro-robotics. However, most of the self-oscillators are based on bending deformation, thereby limiting their possibilities of being implemented in practical applications. Here, we report light-fuelled self-oscillators based on liquid crystal network actuators that can exhibit three basic oscillation modes: bending, twisting and contraction-expansion. We show that a time delay in material response dictates the self-oscillation dynamics, and realize a freestyle self-oscillator that combines numerous oscillation modes simultaneously by adjusting the excitation beam position. The results provide new insights into understanding of self-oscillation phenomenon and offer new designs for future self-propelling micro-robots.

[1] Smart Photonic Materials, Faculty of Engineering and Natural Sciences, Tampere University, P.O. Box 541FI-33101 Tampere, Finland. [2] School of Chemistry and Chemical Engineering, Southeast University, 211189 Nanjing, China. *email: hao.zeng@tuni.fi; arri.priimagi@tuni.fi

Leaves oscillate in response to wind flow. Intriguingly, the frequency of such oscillation can be almost constant, even if driven by a gust of wind that lacks the corresponding periodicity. Leaf oscillation, together with a broad range of phenomena including heartbeat, bridge swaying, sea waves, etc., exemplify naturally occurring self-oscillatory processes[1]. The cause of such self-oscillation is the interplay between the oscillator motion and the force that triggers the oscillation. In the case of leaves, torsional galloping of the leaf shades the aerodynamic vortex generated behind the surface, thus providing positive feedback to the motion that sustains the vibration with a defined periodicity[2]. Generally, the onset of self-oscillation needs to meet critical conditions[3]. This is evident from daily observations: only one or few leaves with specific orientation with respect to the wind flow direction oscillate under a breeze, while the neighbouring leaves with different orientation remain stationary.

Passive oscillating elements in nature rely on an external driving force, such as wind. Stimuli-responsive soft materials allow realizing externally fuelled, man-made self-oscillators, the motion of which is sustained by inner forces that arise due to stimuli-induced changes in material properties. Self-oscillators fuelled by, e.g. light[4,5], heat[6], and chemical reactions[7] have been generated, offering a possibility towards self-sustained motions without the need of human control. Towards this goal, several device functions are demonstrated, including electric power generation[6,8], mass transport[9,10], mill[11], and self-propelling locomotion[5,7,12]. Among the different stimuli, light is particularly promising, due to its sustainability, precise controllability, and omnipresence, the Sun being the ultimate energy source. However, most of the light-driven self-oscillators reported to date are using bending deformation as the main degree of freedom of movement or, in few cases, the combination of bending and twisting[13,14]. The broader the range of available oscillation modes, the more sophisticated autonomous devices one can potentially construct. Therefore, there is a need for self-sustained actuators with versatile motion and complex oscillation modes.

Liquid crystal elastomers and polymer networks (LCNs) are an important class of soft actuators that have been widely used in micro-robotics[15–19]. Their utility is based on coupling between elasticity due to the crosslinked network and anisotropic molecular orientation arising from the LC character. As a result of this combination, LCNs may exhibit pronounced macroscopic shape changes in response to external stimuli. By embedding light-sensitive dyes into the LCN, the material can reversibly and rapidly deform upon light excitation[20]. The photoactuation can be triggered either photothermally[21] or photochemically[22] (or by a combination of these), both leading to light-induced control over the degree of molecular alignment. Programming the molecular alignment in LCNs leads to diverse forms of shape changes[23], enabling the desired deformation mode, and in some cases self-oscillation[4,5,13,14], to be realized.

Typically, light-fuelled self-oscillators rely on a bending LCN strip, in which the oscillation is triggered by alternate activation of the two surfaces[4], or an optical configuration that provides self-shadowing within the motion cycles[24]. Being based on cantilever-type or strip-like geometries, the oscillation has been explained with a model connecting the self-oscillation frequency with their harmonic resonances[24,25]. However, there are some complications in this approach. First, the photomechanical oscillators experience significant damping during the fast motion because of the small mass/inertia compared to the friction/drag. Such damping causes large energy loss and requires compensation from the external stimulus, different to conventional harmonic oscillation. Second, the energy flow into the material by light absorption is oscillating during the cyclic movement, which in turn gives rise to oscillating temperature in the LCN. The change

in intensity/temperature leads to change in modulus[26] and thus to a non-constant resonance frequency.

Owing to the above considerations, several important issues need to be addressed, such as the possibility of obtaining versatile oscillating modes, understanding the feedback mechanism that actually fuels the motion and compensates for the damping loss, and the differences between LCN self-oscillation and conventional cantilever oscillation.

Here we try to address the above questions by exploring the mechanism of self-oscillation in LCN photoactuators. We first theoretically point out that the time delay in the material response provides a positive feedback to the motion, serving as the key to self-sustained oscillation. Then we devise three cantilever-type photoactuators capable of exhibiting the three basic oscillation modes: bending, twisting, and contraction–expansion. We show that the oscillation frequencies of the latter two diverge from the expected natural frequencies. Finally, we demonstrate a freestyle self-oscillator by hanging an LCN photoactuator on a thread, in which all the above degrees of freedom can be combined. The freestyle oscillator has numerous stabilized oscillation modes upon light excitation at different sample positions and is able to evolve spontaneously between different modes upon irradiation with a constant light beam. We believe that the generalized model provided, together with the freestyle self-oscillator demonstrated, can give new insights into soft matter mechanics.

## Results

**Origin of light-fuelled self-oscillation.** While tree leaves oscillate in response to wind flow (Fig. 1a, b), the photoresponse of a soft actuator results from its mechanical deformability upon light illumination. Taking a bending LCN actuator as an example (Fig. 1c), the bending angle increases when increasing the light intensity $I$ (Fig. 1d), which can be qualitatively described as a linear dependence between the deformation ($D$) and the energy absorbed ($E$) by the actuator (Fig. 1f, g). Note that the deformation does not have to be bending as exemplified here but any combination of twisting/bending/contraction, as will be illustrated later. Owing to the directionality of the incident light field, the effective light-absorbing area $A$ depends on the deformed geometry. This can be noticed from Fig. 1e, illustrating that the higher the bending angle, the smaller the light-absorbing area. The relation between $A$ and $D$ can be qualitatively described as a linear curve (Fig. 1g), in which the axis of $A$ can be linked to $E$ through a simple relationship: $E = A \cdot I$. Imposing the two curves gives an intersection, determined by the light intensity, at which the equilibrium occurs (Fig. 1h).

Around this equilibrium, the most familiar case is the harmonic oscillator, where an object experiences a restoring force proportional to its displacement $x$. The equation of motion can be written as:

$$\ddot{x} + \omega_o^2 x = 0, \tag{1}$$

where $\omega_o$ is the angular frequency of the system. Solving Eq. (1) yields harmonic oscillation with a constant amplitude and periodicity $2\pi/\omega_o$, defined by its natural frequency, as shown by the solid line in Fig. 1i. However, in most cases, the oscillator experiences damping from its environment, a restoring force proportional to the velocity of its motion:

$$\ddot{x} + \varsigma \dot{x} + \omega_o^2 x = 0, \tag{2}$$

where $\varsigma$ is the damping ratio. Accounting for the damping, the amplitude gradually decreases, due to energy dissipation from the system to the environment (the dashed line in Fig. 1i).

For a light-responsive mechanical cantilever, the restoring force arises from a sequence of processes: light absorption by the

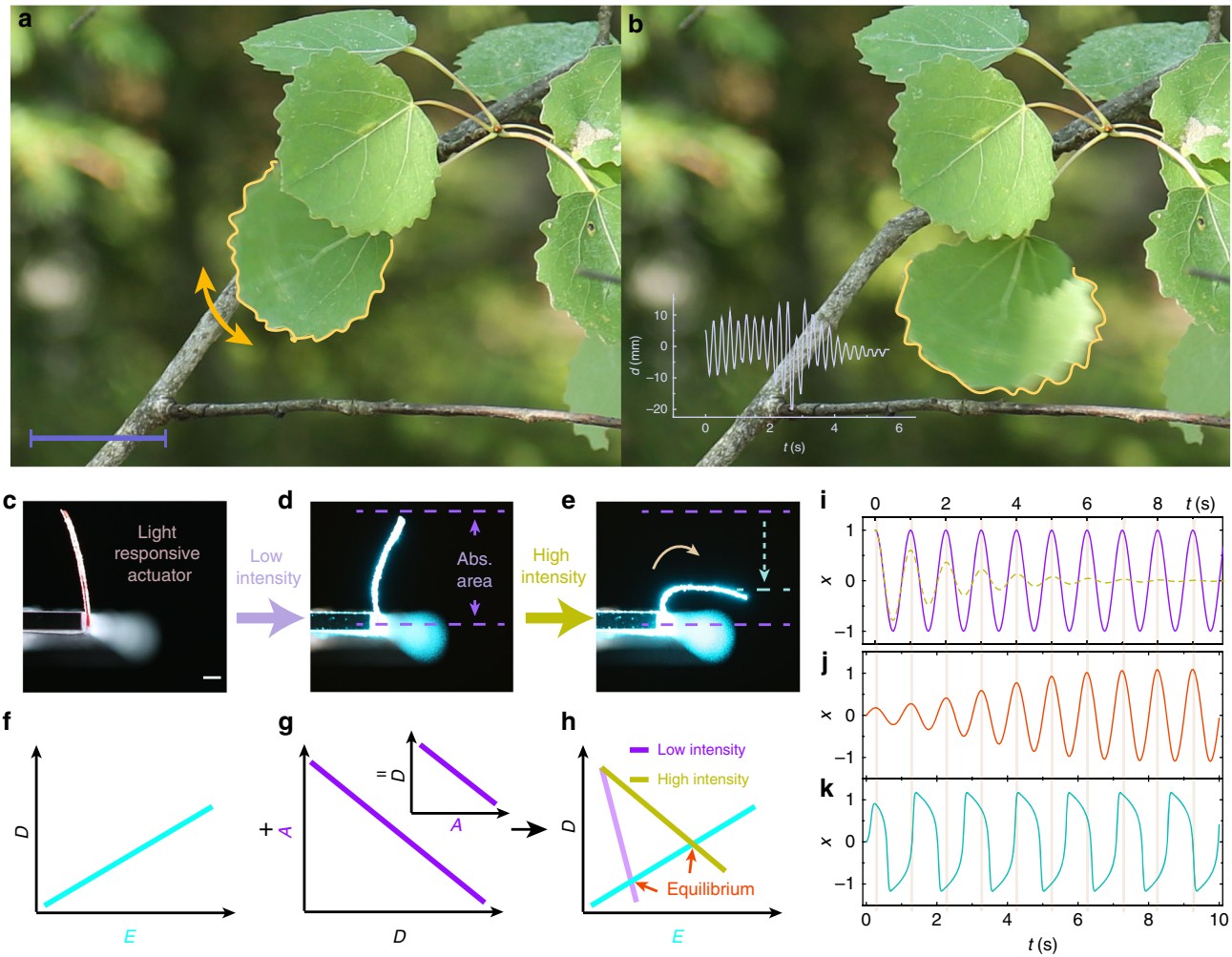

**Fig. 1** The origin of self-oscillation in natural and in artificial light-fuelled systems. **a**, **b** Photographs of a leaf self-oscillating under a breeze. Inset of **b** shows time-dependent displacement of the centre of mass of the leaf, indicating periodic movement with well-defined frequency. Scale bar is 5 cm. A free-standing LCN actuator (**c**) bends slightly upon low-intensity irradiation (50 mW cm$^{-2}$, **d**) and more pronouncedly when the intensity is increased (200 mW cm$^{-2}$, **e**). **f–h** Schematic drawings showing the qualitative dependence between absorbed energy $E$, deformation $D$, effective light-absorbing area $A$, and equilibrium position upon illumination. **i** Harmonic oscillator (solid line, numerical solution to equation, $\ddot{x} + \omega_o^2 x = 0$) and an oscillator experiencing damping (dashed line, numerical solution to equation, $\ddot{x} + \varsigma\dot{x} + \omega_o^2 x = 0$), where $\omega_0 = 2\pi$, $\dot{x}(0) = 0$, $x(0) = 1$, $\varsigma = 0.1$. Self-oscillation induced by **j** a minor time delay $\sigma$ and **k** a large $\sigma$. Numerical solutions to equation $\ddot{x} - (\sigma\omega_o^2 - \varsigma - \eta x^2)\dot{x} + \omega_o^2 x = 0$, where $\varsigma = 0.1$, $\omega_0 = 2\pi$, $\dot{x}(0) = 1$, $x(0) = 0$, and $\sigma\omega_o^2 = 1$, $\eta = 3$ in **j** and $\sigma\omega_o^2 = 20$, $\eta = 60$ in **k**

photoactive molecules, conversion of light energy into heat, heat transport across the sample area, and finally, build-up of the inner stress needed for deformation. All these contribute to a time delay $\sigma$ in the actuation. Thus the equation of motion can be written as

$$\ddot{x}(t) + \varsigma\dot{x}(t) + \omega_o^2 x(t - \sigma) = 0. \qquad (3)$$

The value of $\sigma$ may vary significantly depending on the material system, and it may also depend on the exact position of the LCN within the oscillation cycle. In order to reveal the contribution of this delay into the motion kinetics, we have simplified the model and consider only an invariant $\sigma$ in the motion equation. We use Taylor expansion of Eq. (3), yielding

$$\ddot{x} + \varsigma\dot{x} + \omega_o^2\left(x - \sigma\frac{\dot{x}}{1!} + \sigma^2\frac{\ddot{x}}{2!} - \sigma^2\frac{\dddot{x}}{3!}\cdots\right) = 0 \qquad (4)$$

In this equation, the damping term $-\sigma\omega_o^2\dot{x}$ has a negative value, indicating that the oscillation amplitude increases to a certain extent, until the response becomes nonlinear. For example, air drag would significantly increase at large oscillation amplitudes, leading to amplitude saturation. Herein we adopt a

nonlinear positive damping term $\eta x^2\dot{x}$ (where $\eta$ is a positive constant) based on van der Pol model[27] and simplify Eq. (4) using first-order approximation. The equation of motion for a light-responsive actuator at the equilibrium then appears as

$$\ddot{x} - (\sigma\omega_o^2 - \varsigma - \eta x^2)\dot{x} + \omega_o^2 x = 0. \qquad (5)$$

Numerical solution to Eq. (5) when $\sigma$ is small is given in Fig. 1j, showing an exponential growth of the oscillation amplitude until reaching saturation. Comparing Fig. 1i, j one may notice that all these oscillations exhibit very similar periodicity, matching the conventional cantilever natural frequency, $\omega_o/2\pi$. However, when $\sigma$ increases, the periodicity increases and the form of oscillation becomes non-sinusoidal, as shown in Fig. 1k. Numerical calculations on the oscillation frequency and the waveform upon increasing $\sigma$ are given in Supplementary Fig. 1.

The key to self-oscillation is the time delay in the material response[1]: while an oscillator passes through the equilibrium ($x = 0$), the delayed force remains in the system showing the same direction as the velocity, thus pushing the object out of the equilibrium and providing positive feedback to the motion. Such

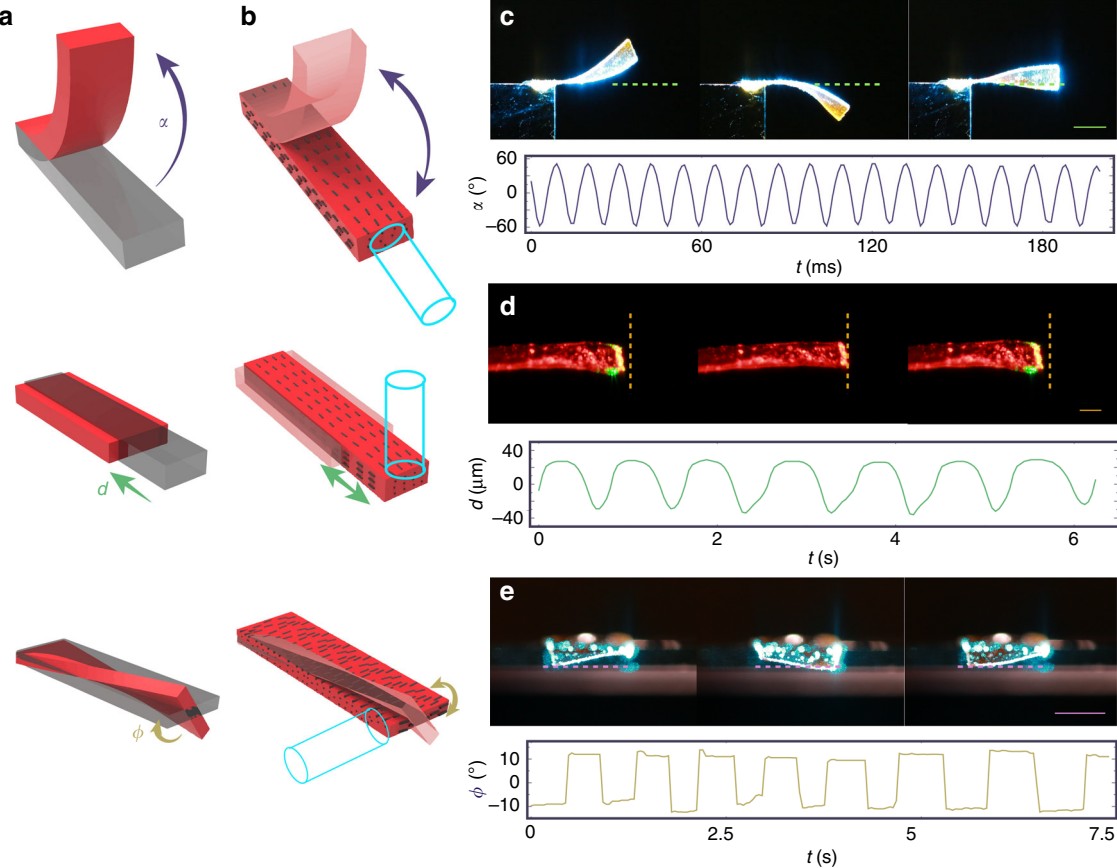

**Fig. 2** Light-fuelled self-oscillators based on three basic deformation modes. **a** Schematic drawing of the three basic deformation modes in a cantilever: bending, contraction, and twisting. The deformation can be quantified through out-of-plane bending angle $\alpha$, displacement $d$, and twisting angle $\Phi$ for each mode. **b** Molecular alignment and irradiation geometry to initiate self-oscillation in each deformation mode. **c** Photographs of bending actuator with size $5.5 \times 1.5 \times 0.05$ mm$^3$ upon 180 mW laser beam (spot size 2.2 mm) and the corresponding oscillation dynamics. **d** Photographs of cylindrical actuator (100 μm diameter) upon 100 mW focused laser excitation (spot size 20 μm) and the contracting–expanding oscillation dynamics. **e** Twisting self-oscillator (size: $3.5 \times 3.5 \times 0.05$ mm$^3$, laser: 180 mW, spot 2.2 mm) and the corresponding oscillation dynamics. The scale bars are 2 mm in **c**, **e**, 100 μm in **d**

feedback extracts the energy from the external energy source (light), fuelling the oscillator to sustain the motion. In the following, we explore the relation between the deformation degrees of freedom and the equilibrium positions in different light-fuelled self-oscillators and connect the observed oscillation modes to the material delay.

**Basic oscillation modes**. The three basic deformation modes in a cantilever—bending, contraction–expansion, and twisting—are schematically shown in Fig. 2a. Their amplitudes can be quantified as out-of-plane bending angle $\alpha$, displacement $d$, and twisting angle $\Phi$ around the cantilever axis, respectively. To realize the basic deformations, we used planar-aligned LCN cantilevers, with a director along the cantilever for bending and contraction and perpendicular to it for twisting (Fig. 2b). The bending/twisting and contracting actuators are based on different LCN materials, as further elaborated in "Methods" and the Supplementary Methods.

The photoactuation is triggered with a 488 nm continuous-wave laser beam, using different irradiation geometries for the different modes as depicted in Fig. 2b. Bending is obtained by light propagation along the cantilever axis, in which case the laser alternately activates both surfaces during the cyclic motion. The photoactuation results from light absorption gradient across the thickness, which in turn gives rise to photoinduced stress on the irradiated surface, and bending towards the illuminating light

source. Combination of photochemical and photothermal effects is likely responsible for this type of oscillation[13]. Contraction occurs when the beam is focused on the tip of the cantilever, heating up the material, followed by subsequent cooling and expansion once the cantilever has moved away from the laser spot. Thus photothermal effects are responsible for this actuation mode. Twisting/torsional deformation takes place when the strip is illuminated from the side, causing an angular displacement around the cantilever axis and alternate exposure of the two cantilever surfaces, driven by similar mechanism as in the case of bending. Further details on photoactuation and the experimental arrangements are given in "Methods" and Supplementary Methods, respectively.

Equilibrium positions, where $\alpha = d = \Phi = 0$, can be found for each mode. When driven out of equilibrium (initiated by air fluctuation or a mechanical trigger), the cantilever returns to the equilibrium position due to elastic force, and the delay in the material response causes the self-oscillation as explained earlier. The frequency of the self-oscillation depends on the desired mode. The bending oscillation shown in Fig. 2c has the highest frequency of 80 Hz among the three basic modes (see the slow-motion video, Supplementary Movie 1), well matching with the calculated natural resonance frequency of the cantilever (about 90 Hz, see natural frequency calculations in Supplementary Methods for further details). This indicates a minimal time delay $\sigma$ in such mode, yielding self-oscillation dominated by $\omega_o$ (Eq. (5), Fig. 1j). The natural resonance frequency is determined by

the cantilever geometry and the material rigidity, as has been systematically studied in several LCN self-oscillators[24,25]. In contrast, the contraction–expansion oscillator as well as the twisting one exhibit much longer periodicities, and resonance frequencies in the range of 0.5–18 Hz for contraction and about 1 Hz for twisting can be observed (Fig. 2d, e; Supplementary Movies 2 and 3). By comparing the observed frequencies to the natural resonances, estimated to be >20 kHz for the longitudinal vibration mode and ca. 170 Hz for the torsional one (see natural frequency calculations in Supplementary Methods), the frequencies of light-fuelled self-oscillations are 2–4 orders of magnitude lower. We ascribe this to significant increase in $\sigma$ in those modes. During contraction–expansion, the light energy is absorbed within the focused spot area located at the tip of the cantilever. After the light absorption, heat has to be conducted along the cantilever axis to trigger deformation, thus posing a huge time delay. This delay strongly depends on the amount of energy being absorbed by the material. A set of stable oscillations with a broad range of frequencies, from 0.5 to 18 Hz, are observed by slightly tuning the laser spot position (same power) on the sample (Supplementary Fig. 2). We attribute this to differences in light scattering due to inhomogeneity and hence different amount of light being absorbed, which in turn yields different contraction length and time delay in the material response. In the twisting-mode oscillator, the time delay arises from the stress accumulation required for the material to switch its angular orientation. In this case, there seem to be two stable states, as indicated by the rectangular waveform shown in Fig. 2e.

More detailed analysis on the contraction–expansion and twisting self-oscillation modes have been conducted, with results shown in Supplementary Figs. 3 and 4. By maintaining the same excitation beam position but changing the input power, the frequency of the contraction–expansion mode varies between 16 and 20 Hz (see the Fourier transform in Supplementary Fig. 3b). However, no direct relation between the oscillation frequency and the excitation power can be deduced. The twisting actuator, instead, maintains the same oscillation amplitude irrespective of the laser power, but its oscillation frequency increases from 0.7 to 2 Hz (Supplementary Fig. 4b, c). Such findings seem to contradict conventional conceptions about cantilever mechanics, where the structural rigidity determines the natural resonance frequency, while the input power may affect only the oscillation amplitude. These observations, together with the non-sinusoidal waveforms and the considerably low oscillation frequencies compared to the natural resonance frequencies, indicate that the time delay becomes the dominating factor in the contracting and twisting self-oscillators. Note that, for large time delays, the simplified model given by Eq. (5) will no longer be accurate and more elaborate numerical calculation methods would be needed for solving the corresponding kinetic equations.

**A freestyle self-oscillator**. The actuators described above exhibit only one selected oscillation mode. In order to combine the different modes into a single actuator, we use a main-chain LCN with high degree of deformability[28] (>50% contraction upon order–disorder phase transition; thermal actuation and mechanical properties between different LCNs given in Supplementary Figs 5 and 6). Material details are given in "Methods"; for further information about sample preparation, see the Supplementary Methods. Upon excitation with a laser spot, the localized strain builds up stress inside the material, causing additional degrees of freedom for the entire deformed structure, as schematized in Fig. 3a and experimentally demonstrated in Fig. 3b–e. The strip (Fig. 3b) bends when illuminated with a laser spot (100 mW) at the centre of the cantilever (Fig. 3c). Moving the laser spot

horizontally, close to the edge of the cantilever, triggers combined twisting and bending deformation (Fig. 3d). By increasing the light intensity from 100 to 140 mW, a larger extent of deformation can be triggered, and bending and contraction occur simultaneously (Fig. 3e). To better visualize the twisting, we attached a weight to the end of the actuator (1.4 g, 250 times the weight of the actuator) to suppress bending (Fig. 2f). When the laser spot deviates from the centre of the actuator, contraction and twisting can be observed (Fig. 2g).

To trigger self-oscillation, the excitation beam orientation and spot location need to meet critical conditions. We realize this by puncturing a hole into the LCN (inset of Fig. 3j) and hanging it vertically on a thread (human hair), thereby providing freedom to swing around the thread axis, as schematically illustrated in Fig. 3h. Under such configuration, any kind of light-induced deformation (bending and/or contraction and/or twisting) shifts the centre of mass of the actuator (Fig. 3h; $1 \rightarrow 2$). In return, gravity imposes a torque to rotate the deformed structure around the thread axis ($2 \rightarrow 3$), and the strip moves out of the illuminating area, cools down, and relaxes back to the original position ($3 \rightarrow 1'$), and the cycle restarts.

Triggered by subsequent sequences of the above-described cycles, the strip oscillates continuously upon a constant laser beam excitation. By slightly changing the position of the laser spot, different oscillation modes can be obtained (Fig. 3i–k and Supplementary Movie 4). Each mode has a well-defined periodicity, as exemplified in the inset of Fig. 3k. During the distinct stable oscillation modes, the laser beam is not spatially or temporally modified. However, by slightly adjusting the laser spot position along the LCN strip, the actuator may exhibit fast-varying geometry, oscillating like a dancer on a thread (Supplementary Movie 5), which we coin as light-fuelled freestyle self-oscillation.

**Stabilized oscillations and evolution between the modes**. For more detailed characterization of the oscillating modes, we labelled the free end of the LCN strip with fluorescent particles (Fig. 4a, and further details are given in data collection in Supplementary Methods). With the help of the fluorescent markers, we can track the trajectory of oscillation in the $x$–$y$ plane, offering a distinct fingerprint for each oscillation mode. Each mode prevails for a relatively long time, typically at least 5 s, maintaining the same amplitude and frequency (Supplementary Fig. 7). However, the oscillation frequency may significantly vary for the different modes, e.g. from 0.4 to 6.7 Hz between examples shown in Fig. 4b–g, and no connection between the oscillation frequency and amplitude seems to exist. The trajectories with zero area correspond to reciprocal oscillation (Fig. 4e–g) while those with non-zero area indicate non-reciprocal motion during the oscillation cycle (Fig. 4b–d).

Figure 4h presents an intriguing oscillation behaviour under constant irradiation conditions: spontaneous evolution between different oscillation modes. In short term, the periodicity of each mode is constant (Fig. 4k) but varies somewhat over extended time periods, e.g. from 0.35 to 0.45 s within 3 min (Supplementary Fig. 8). The spontaneous evolution between the modes (Fig. 4i, j) without changing the excitation beam position is ascribed to irreversibility of the actuating material, which is quantified by monitoring a freestanding LCN upon cyclic excitation (Supplementary Fig. 9). The cyclic actuation shows that the material maintains its deformability, i.e. the maximum strain obtained upon constant irradiation, while the relaxation process shows certain degree of reduction with a decay constant of about 80 cycles. The reason for such photomechanical irreversibility may be attributed to kinetic deformation driven

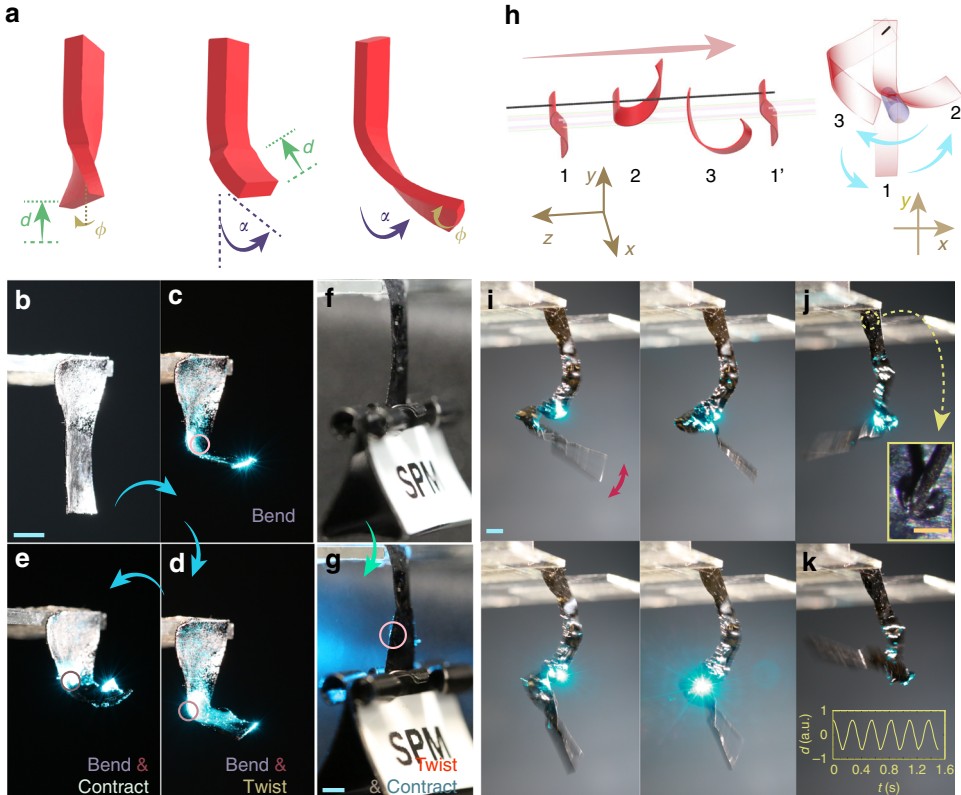

**Fig. 3** A freestyle self-oscillator. **a** Schematic drawing of a soft actuator combining the three basic deformation modes (twisting, bending, and contraction). An LCN actuator (**b**) that bends (**c**), bends and twists (**d**), bends and contracts (**e**), upon changing the irradiation conditions. The circle marks the position of the laser spot. With an external load that prevents bending (**f**), the actuator contracts and twists simultaneously when excited from one side as indicated in **g**. **h** Schematics of a light-driven self-oscillator swinging on a thread. **i–k** Snapshots of the freestyle oscillator with different oscillation modes upon laser excitation (488 nm, 160 mW) at different positions. LCN strip size: 3 × 25 × 0.1 mm$^3$. Inset of **j** shows the microscopic image of the hole punctured in the LCN actuator (scale bar 200 μm). Inset of **k** shows the displacement of the end of the strip during the oscillation process. All scale bars: 2 mm

by elastic entropy, where cyclic stretching–coiling gives rise to friction between the polymeric chains, and photobleaching of the dyes. During the oscillation, the actuator undergoes hundreds of deformation–relaxation cycles. Hence, even slight changes in material response may shift the deformation mode and lead to spontaneous evolution between the oscillation modes. At about 190 s, the oscillator stops at a stable position, where the negative damping condition is no longer met ($\sigma\omega_o^2 > \varsigma$, Eq. 5), possibility due to the irreversibility of the material response.

The prediction and control of the evolution between different modes during the freestyle oscillation is difficult to attain, due to the multiple degrees of freedom in deformation and complexity of light–matter interactions in responsive soft materials, resulting in different time delay in material response at different stages of the oscillation cycle. Despite the lack of control, the intriguing freestyle self-oscillation phenomenon provides insight into the process that, we believe, can be generalized into different kinds of responsive materials with arbitrary geometry, deformability, and stimulus responsivity, as long as a feedback mechanism has been established for sustaining the periodic motion.

**From general material concept towards practical applications**. The above light-driven system combines the basic oscillation modes—bending, contraction, and twisting—into a single free-style oscillator. Although the movements are complex, the free-style oscillator is governed by the same physics as conventional oscillators with simpler movements. We believe that similar principles can be applied also to other types of stimuli-responsive

systems. For example, thermoresponsive nylon fibres[9], carbon-nanotube-based bi-layer actuators[29], thermoactive elastomer rods[10], hydrogels[30], and azobenzene crystals[31] have all shown self-sustained cyclic movements. Different from approaches in systems chemistry[32,33], which aim at programming positive/negative feedback between reversible chemical reactions to induce oscillating behaviours, the approach demonstrated here focuses on photomechanical response in macroscopic samples, driven by time delay in material response and not restricted in few specific materials. Also note that the self-oscillation discussed in this study is different from the phenomenon of fluctuation, which, however, in some cases has been coined as chaotic self-oscillation[34]. A well-defined frequency component in the Four-ier transform can serve as a good indicator for the occurrence of self-oscillation.

To combine self-oscillators into locomotive devices to achieve self-sustained micro-robots is challenging. First, most of the actuators used in self-oscillation studies are free-standing strips lacking external loading. Upon devising a linkage between the motor (the oscillator) and the robot body (the moving element), the mechanical conditions change. Upon locomotion, the oscillator's position with respect to the illumination source, as well as its geometry and mechanical properties, change, affecting the equilibrium conditions crucial for sustaining the oscillation. Thus a conventional mechanical design, e.g. biking the wheel[35], may lead to a failure in long-time conversion of oscillating motion into mechanical work. Second, efficient translocation often requires a nonreciprocal deformation, which in turn requires temporal coordination of a series of self-oscillation

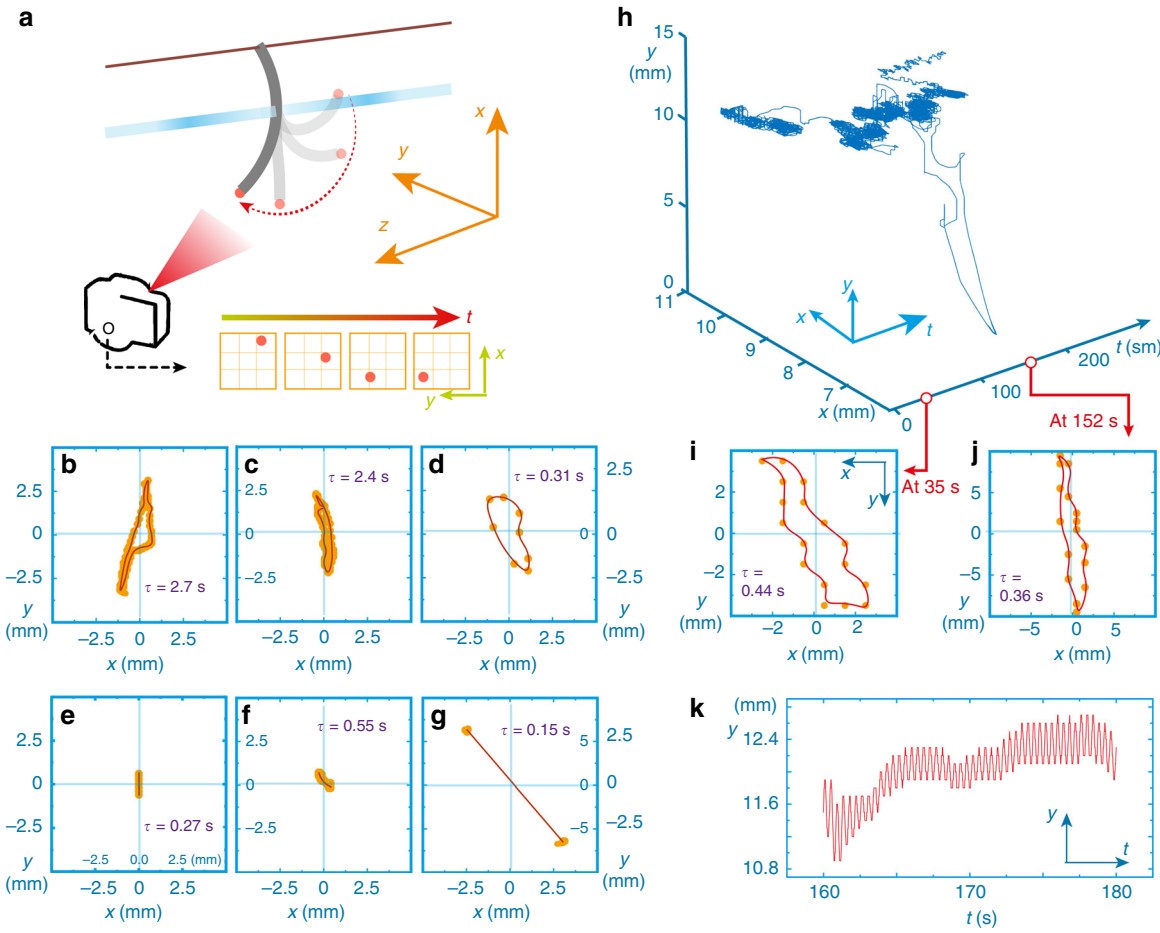

**Fig. 4** Stabilized modes and evolution between the modes. **a** Schematic drawing of the arrangement for tracking the oscillator position. **b**–**g** Examples of different self-oscillation modes upon excitation with a laser spot at different positions. Data shows the tracking trajectory on *x*–*y* plane for one oscillation cycle. **h** Evolution between modes upon a constant laser excitation. The *x*–*y* trajectory spontaneously changes upon prolonged excitation (**i**, **j**), while maintaining the similar oscillating periodicity (**k**) until 190 s when the oscillation vanishes

events or within different sequences of a self-oscillation cycle with certain phase delays[36,37]. However, coupling between different self-oscillators sets extra hurdles. Third, light-responsive materials may not provide enough force to overcome the friction/drag experienced during the robotic motion, hence other soft mechanic properties like snapping and instabilities[37] need to be further investigated in self-oscillating systems, in order to reach their full potential. Although challenges exist, pioneering examples have shed light on device integration, e.g. mass transportation[9,10] and self-propelled walking[5]. We hope more self-sustained devices will be reported in the future, and that self-sustained robots can be integrated with more sophisticated functions by the joint scientific development in this field.

A useful application can be foreseen in realizing self-oscillator within fluidic environment, which may eventually give birth to self-sustained pumping and cilia-like collective movement for efficient microfluidic manipulation. However, the dragging force (large $\varsigma$) may pose great hurdle to meet the required negative damping condition ($\sigma\omega_o^2 > \varsigma$). A possible solution could be to investigate stimuli-responsive materials with a large delay $\sigma$ in material response, such as light/heat-responsive hydrogels[38].

Beyond the above-mentioned robotic applications that aim to transfer input light energy into mechanical work output, there is a frontier of optical applications to be investigated. A self-oscillator is essentially an optical chopper. However, compared to the conventional electronic motorized chopping system, it has many

advantages, such as compact size, light weight, and, being remotely powered, possibility towards integration into micro-devices. An oscillating strip can modulate a signal beam (635 nm) periodically in its transmission, as shown in Fig. 5a: during the down-bending stage, the beam is transmitted, while the up-bending stage blocks the beam. In other words, the signal beam transmission is controlled and modulated by the self-oscillator (Fig. 5b). Furthermore, one single strip can even control two signal beams simultaneously. As shown in Fig. 5c, the two signal beams can be in phase, when propagating along the same side of the strip, or 180° out-of-phase, when propagating from the opposite sides of the strip (Fig. 5d).

## Discussion

We study the mechanism of light-driven self-oscillation and show that a time delay due to the material response is the key to sustain the periodicity of oscillation. By interplay between the equilibrium position upon irradiation and specific form of light-induced deformation pre-programmed into the LCN by molecular-alignment control, three basic oscillation modes—bending, contraction, and twisting—are demonstrated. A main-chain LCN actuator combines the basic oscillation modes, yielding freestyle oscillation when hung on a thread. Such freestyle oscillator exhibits different stabilized modes upon continuous laser excitation at different sample positions and evolution between the modes across a relatively long period under

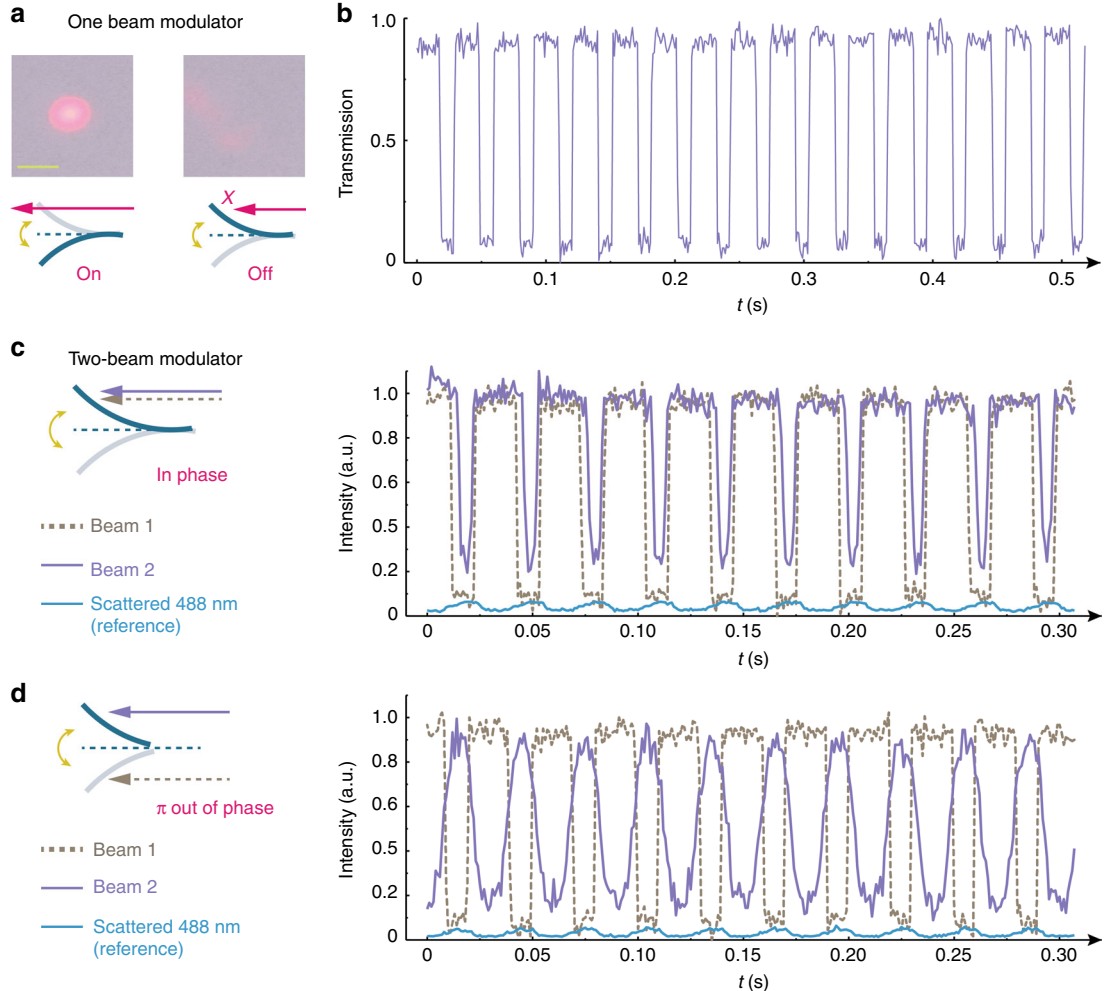

**Fig. 5** Self-oscillator-based beam modulator. **a** A self-oscillating strip (bending mode, driven by a continuous 488 nm, 190 mW) can modulate the transmission of a signal beam (635 nm). The signal beam is focused on the upper part of the oscillator as shown by the schematic drawing, while the transmitted beam profiles are shown in the photos. Scale bar: 1 cm. **b** Transmission measurement of the beam being modulated by the self-oscillator. **c** Two signal beams are propagating along the same side of the oscillator and the measured transmission intensities being modulated in phase. **d** Two signal beams are propagating along the opposite sides of the oscillator and the measured transmission intensities being modulated 180° out-of-phase. Blue curves in **c**, **d** show the scattered 488 nm light intensity for phase reference. Strip size: $1 \times 10 \times 0.05$ mm$^3$

a constant light field. The results provide insight into light-driven self-oscillation in soft materials and may in longer term open up new approaches to realize light-driven motors for self-sustained soft micro-robotics.

## Methods

**Materials in brief**. The LCN photoactuators demonstrated here are photo-polymerized from monomer mixtures containing liquid crystalline mesogens, crosslinkers, light-sensitive dyes, and photoinitiators, via one-step or two-step fabrication strategies. Different materials were used for different deformation/oscillation modes. For bending and twisting, a side-chain LCN with 28 mol% crosslinker (of which 6 mol% are azobenzene crosslinkers) is used[39]; for contraction–expansion, we used a side-chain LCN polymerized from different LC mesogens and reduced crosslinker concentration (10 mol%)[40] to render the LCN softer; for freestyle actuator, we used a main-chain LCN synthesized through two-step fabrication process[28]. All details about material preparation procedures are given in Supplementary Methods.

**Photoactuation in brief**. An unfocused, continuous-wave 488 nm laser beam was propagated along the axis of the LCN cantilever in case of bending-mode oscillation and perpendicular to the axis for the twisting mode. Upon laser excitation, the azobenzene moieties in the LCN undergo repeated *trans-cis-trans* isomerization. Owing to limited light-penetration depth, isomerization gradient across the film thickness is formed, which, together with photothermal heating, gives rise to

cantilever deformation towards the light source. As a result, the laser beam illuminates alternately both cantilever surfaces, yielding periodic bending/twisting. Further details on the optical configuration are given in Supplementary Methods. For contraction–expansion mode oscillation, we used a purely photothermally driven actuator, where light was absorbed at a tip of an LCN fibre and the energy was transferred into heat and conducted along the cantilever to the rest of the actuator body. The temperature difference between illuminated and non-illuminated surfaces is negligible due to fast thermal transport occurring at microscopic scale. Bending and twisting actuators are film like, with length-to-thickness ratio about 100. For contraction–expansion mode, a microscopic fibre with 100 μm diameter was used, with length-to-thickness ratio around 1. Because of the low length-to-thickness ratio, the bending of the fibre was suppressed. The LCN used for freestyle oscillation was also a photothermal actuator. Because of its large deformability, multiple degrees of freedom of deformation can be activated by impinging the light beam at different sample positions and/or by changing the excitation power. In all cases, the excitation beam position needs to be finely tuned in order to reach the stable self-oscillating movement. Once it self-oscillates, the duration can last for typically >30 min for bending and twisting modes, >50 min for contacting–expanding mode, and >30 s for the freestyle mode. More details about photomechanics and optical configurations are given in Supplementary Methods.

## Data availability

The data that support the findings of this study are available from the corresponding authors upon request.

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

## Acknowledgements

The work is part of the Academy of Finland Flagship Programme, Photonics Research and Innovation (PREIN), decision number 320165. This work is supported by ERC (Starting Grant PHOTOTUNE, Agreement No. 679646) and Academy of Finland postdoctoral grant (Decision no. 316416 & no. 326445). H.Y. thanks Jiangsu Provincial Natural Science Foundation of China (BK20170024) for financial support. We are indebted to A. Berdin for assistance with Young's modulus measurements and A. Khan for MATLAB coding. Dr. P. Wasylczyk (Warsaw University), Dr. H. Zhang and Pro-fessor O. Ikkala from Aalto University are acknowledged for inspiring discussions and insightful comments.

## Author contributions

H.Z. and A.P. conceived the project. H.Z. carried out experiments with the help of M.L. and O.M.W. M.L. prepared LCN for bending and twisting oscillation mode. Z.A. syn-thesized side-chain LCN for contracting oscillation mode. L.L. synthesized the main-chain LCN under supervision of M.W. and H.Y. All authors contributed in writing the manuscript.

## Competing interests

The authors declare no competing interests.
