## [Peer Review File · Nature Communications]

Reviewers' comments:

Reviewer #1 (Remarks to the Author):

The manuscript of Zeng et al from the Priimagi group describes the principles for light-fuelled self-oscillating systems on the basis of liquid crystal elastomers. On itself is the self-oscillation of thin films or beams made from liquid crystal elastomers well-known, and the corresponding literature places are widely cited by the authors. In addition to what is known, the authors' objective with their research complemented in the manuscript is, as I interpreted, to treat and discuss the oscillatory principles in more general perspective. Thereto they studied different types of motion figures, such as bending (widely described in literature), twist and elongational expansion/shrinkage. The latter two are less common in literature and I especially liked the elongational effects which were new to me. In addition, they provide mathematical support including geometrical aspects of the changing light spot and terms for feedback and damping. Despite the still relatively simple approach they were able to describe the dynamics of oscillation and explain whether the oscillation is sustained or damped.

With respect to recommendation for publication I am somewhat in doubt. On the positive side I liked the approach to generate more general and broader description of the oscillatory phenomena than presently described in literature. On the negative side, I am somewhat disappointed in the sample set the authors prepared and presented in the related figures added to the manuscript. The group is well-known for their usually well-engineered responsive devices with a high degree of perfection. However, in this manuscript the demonstrators, especially the ones in the figure of the manuscript, look rather poor. On the other hand in some of the videos the predicted effects of oscillation are clear. And in that respect 209685_0_related_ms_490843_pqz2rz is rather spectacular. But the other movies around the tread-mounted films, which should support the addition 'free-style' to the title of the publication, provide random deformations where gravity and created momentum play a unpredicted role. To my opinion this part can be left out of the publication and more emphasis should be put on the basic deformation figures and the quantification of their effects, including a better analysis of the conditions (light intensity, temperature, temperature gradients, etc.) when oscillation becomes sustained.

Reviewer #2 (Remarks to the Author):

See attached report.

The paper is novel and it attacks a complex and interesting problem. I should be published. I have only minor comment (see attached report).

Reviewer #3 (Remarks to the Author):

Liquid crystal network (LCN) material is a one of the widely investigated intelligent actuation materials. Recently, lots of important studies have been carried out on the self-oscillating motion of LCN (e.g. Nat. Commun., 2016, 7, 11975; Nature, 2017, 546, 632.). This work proposes that different self-oscillation modes can be realized by illuminating different positions of the different materials, but lacks the experimental results or at least some prototype devices for demonstration. Therefore, I suggest the author to make further investigation and modification.

The comments are shown as follows:

1. In Figure 2d (contraction mode), light is incident from the upper perpendicular to the upper surface, so the lower surface is not illuminated. Does this difference in light irradiation between the upper and lower surfaces lead to some bending deformations?

2. In page 13, line 247, "The strip (Fig. 3b, i) bends when illuminated with a laser spot (100 mW) at the centre of the cantilever (Fig. 3b, ii)." This way for realizing bending mode is different from the description shown in figure 2 "Bending is obtained by light propagation along the cantilever axis, in which case the laser alternately activates both surfaces during the cyclic motion." Please give the reason.

3. In page 12 line 250, the authors mentioned that "By increasing the light intensity from 100 to 140 mW, a larger extent of deformation can be triggered, and bending and contraction occur simultaneously (Fig. 3b, iv)." Without changing the position of illumination, the oscillation mode can be change only be increasing the light intensity? It's different from the previous description of the contraction mode, why?

4. The authors explain that the final stop of the self-oscillation is resulted from the irreversibility of materials response. The more specific explanations should be provided. What about the other oscillation such as bending, contraction and twisting? Do they finally also reach the stable position under the constant light illumination?

5. In figure 2, different oscillation modes were realized by using different LCN materials with different dimensions. During this self-oscillation process, the laser beam is illuminated on the different position of different materials, and the spot size of the laser beam are all different. In contrast, in figure 3, the different oscillation modes are achieved by only changing the position of the laser spot on one same sample. The more detailed analysis and explanation must be provided.

6. Does the evolution of the self-oscillation can be stably modulated from a mode to another certain mode?

7. The application examples of the self-oscillators with different oscillation modes should be given to demonstrate their practical application prospect.

Answers to the Reviewers' comments

Reviewer #1

The manuscript of Zeng et al from the Priimagi group describes the principles for light-fuelled self-oscillating systems on the basis of liquid crystal elastomers. On itself is the self-oscillation of thin films or beams made from liquid crystal elastomers well-known, and the corresponding literature places are widely cited by the authors. In addition to what is known, the authors' objective with their research complemented in the manuscript is, as I interpreted, to treat and discuss the oscillatory principles in more general perspective. Thereto they studied different types of motion figures, such as bending (widely described in literature), twist and elongational expansion/shrinkage. The latter two are less common in literature and I especially liked the elongational effects which were new to me. In addition, they provide mathematical support including geometrical aspects of the changing light spot and terms for feedback and damping. Despite the still relatively simple approach they were able to describe the dynamics of oscillation and explain whether the oscillation is sustained or damped.

We thank Reviewer for his/her comments about the novelty of our findings.

With respect to recommendation for publication I am somewhat in doubt. On the positive side I liked the approach to generate more general and broader description of the oscillatory phenomena than presently described in literature. On the negative side, I am somewhat disappointed in the sample set the authors prepared and presented in the related figures added to the manuscript. The group is well-known for their usually well-engineered responsive devices with a high degree of perfection. However, in this manuscript the demonstrators, especially the ones in the figure of the manuscript, look rather poor. On the other hand in some of the videos the predicted effects of oscillation are clear. And in that respect 209685_0_related_ms_490843_pqz2rz is rather spectacular. But the other movies around the tread-mounted films, which should support the addition 'free-style' to the title of the publication, provide random deformations where gravity and created momentum play a unpredicted role. To my opinion this part can be left out of the publication and more emphasis should be put on the basic deformation figures and the quantification of their effects, including a better analysis of the conditions (light intensity, temperature, temperature gradients, etc.) when oscillation becomes sustained.

We are grateful for the reviewer for the constructive criticism, motivated by which many additional details on light-induced self-oscillations were added to the revised manuscript. What we would like to highlight is that this manuscript aims to provide new insights into the general phenomenon of light-fuelled self-oscillation. To do this, we constructed a simple model to describe the phenomenon, and experimentally demonstrate several forms of self-oscillation and ultimately, a free-form self-oscillator that combines several oscillation modes. Our aim is not to present well-engineered devices or actuators, but rather to try to elaborate that any active material, no matter how irregular in shape or versatile in deformation, can self-oscillate under a constant light beam. Towards this goal, each figure in the manuscript plays an important role. For instance, Figs. 2c, 2d, and 2e present three LCN strips exhibiting bending, contracting-expanding and twisting modes; Figs. 3b, 3c, and 3e demonstrate that an irregular LCN strip combines the three basic deformation modes shown in Fig. 2, and presents the freestyle self-oscillator that is further studied in Fig. 4. Although the freestyle self-oscillator lacks controllability, we believe that the free form and random, almost chaotic, motions well represent the meaning of "freestyle".

We would like to re-emphasize that the main finding of this manuscript is an LCN strip with irregular shape, free deformation form and multiple degrees of deformation freedom, which can still self-oscillate under a non-modulated, continuous laser beam. This intriguing phenomenon, we believe, will provide insights into the field of soft mechanics (i.e., beyond LCNs), and thus attract broad readership. Hence, we see it important to keep that part in the publication.

As for the controllability and predictability in the self-oscillating modes, according to Reviewer's suggestion, we have provided detailed analysis on the self-oscillating properties of twisting and contracting-expanding modes, which have been much less studied than the bending mode. The below text has been added to the manuscript, and two figures into the Supplementary Information.

More detailed analysis on the contraction-expansion and twisting self-oscillation modes have been conducted, with results shown in Supplementary Figs. 3, and 4. By maintaining the same excitation beam position but changing the input power, the frequency of the contraction-expansion mode varies between 16 and 20 Hz (see the Fourier transform in Supplementary Fig. 3b). However, no direct relation between the oscillation frequency and the excitation power can be deduced. The twisting actuator, instead, maintains the same oscillation amplitude irrespective of the laser power, but its oscillation frequency increases from 0.7 to 2 Hz (Supplementary Fig. 4 b,c). Such findings seem to contradict conventional conceptions about cantilever mechanics, where the structural rigidity determines the natural resonance frequency, while the input power may affect only the oscillation amplitude. These observations, together with the non-sinusoidal waveforms and the considerably low oscillation frequencies compared to the natural resonance frequencies, indicate that the time delay becomes the dominating factor in the contracting and twisting self-oscillators. Note that for large time delays, the simplified model given by Eq. (5) will no longer be accurate and more elaborate numerical calculation methods would be needed for solving the corresponding kinetic equations.

Supplementary Figure 3 | Frequency analysis of an LCN cantilever with contraction-expansion -type oscillation mode. (a) A typical self-oscillation event recorded by measuring the scattered light intensity from the tip of the LCN during the oscillation. Laser power: 125 mW. A zoom-in image is given in (c). **(b)** Fourier transform at the frequency domain for self-oscillations upon identical excitation beam position but varying the laser power. **(d)** Evolution of self-oscillation frequency

along with increase of laser power. The fiber-like LCN is about 100 μm in diameter, and excited with a 488 nm laser beam focused by 10 \times objective (NA: 0.25).

Supplementary Figure 4 | Frequency analysis on an LCN cantilever with twisting mode. (a) A series of photographs showing cyclic twisting oscillation, with arrows indicating the periodicity τ of each oscillating cycle. **(b)** Change of oscillating periodicity upon irradiating with different laser power. Error bars indicate standard deviation of $n > 10$ oscillation periods. **(c)** The oscillation amplitude remains constant with increasing laser power. The size of the LCN is of $3 \times 4.5 \times 0.05 \text{ mm}^3$, and it is excited with a non-focused 488 nm laser beam.

Reviewer #2

The paper is novel and it attacks a complex and interesting problem. It should be published. I have only minor comment (see attached report). This interesting paper models how light can drive solid oscillators cyclically, even if the light is not varying. The geometry of the distortions themselves give the variations of drive that are required, so these are interesting phenomena. The route the authors take is translating these effects into a temporal delay that gives a van der Pol type drive. I only have a few small comments and questions:

We would like to thank the Reviewer for his/her positive comments.

1. The word ‘perpetual’ is used, but in these contexts the word has unfortunate connotations (“perpetual motion”). These are entirely unintended by the authors and they would be well-advised to find another turn of phrase! (See also later in the paper at line 329.)

To avoid potential misunderstanding, we have removed the word “perpetual” from the manuscript, as suggested.

2. The time-delay introduced in (3) is the key. It seems the simplest assumption. I wonder whether the authors would like to comment on the delay being independent of where the oscillator is in its cycle (ie notions of self-eclipsing etc.)?

We admit that the model we provided is a very simple one. It only shows the most important aspect behind self-oscillation, which is the time delay in the material response. Application to specific cases of our demonstrated oscillators is challenging, since the material delay indeed is cycle-/position-dependent. We have added a short discussion related to this to the revised manuscript.

The value of σ may vary significantly depending on the material system, and it may also depend on the exact position of the LCN within the oscillation cycle. In order to reveal the contribution of the delay into the motion kinetics, we have simplified the model, and consider only an invariant σ in the motion equation.

3. Line 146: where did the non-linear positive damping term come from? (A small motivation would be good.)

We believe the non-linear damping to be due to air drag, which would significantly increase upon increasing oscillation amplitude. We have added the following text to the manuscript to address this:

In this equation, the damping term $-\sigma\omega_0^2\dot{x}$ has a negative value, indicating that the oscillation amplitude increases to a certain extent, until the response becomes nonlinear. For example, air drag would significantly increase at large oscillation amplitudes, leading to amplitude saturation. Herein, we adopt a nonlinear positive damping term $\eta x^2\dot{x}$ (where η is a positive constant) based on van der Pol model²⁷, and simplify Eq. (4) using 1st-order approximation.

4. Line 213: *I did wonder at this point whether inertia terms in the equation of motion are really significant or not (x'')?*

The question whether $\sigma \ll 1$ is a significant one, and it actually points out the limitation of the simplified model we use for explaining the self-oscillating phenomenon. For self-oscillation with significant delay, equation (5) will no longer be accurate. Instead, numeric calculation methods can be adopted in modeling the response for such cases. We have added the following related discussion to the text:

Note that for large time delays, the simplified model given by Eq. (5) will no longer be accurate and more elaborate numerical calculation methods would be needed for solving the corresponding kinetic equations.

Reviewer #3

Liquid crystal network (LCN) material is a one of the widely investigated intelligent actuation materials. Recently, lots of important studies have been carried out on the self-oscillating motion of LCN (e.g. Nat. Commun., 2016, 7, 11975; Nature, 2017, 546, 632.). This work proposes that different self-oscillation modes can be realized by illuminating different positions of the different materials, but lacks the experimental results or at least some prototype devices for demonstration. Therefore, I suggest the author to make further investigation and modification.

We thank reviewer for his/her suggestions. Before addressing them point by point, we briefly address the two articles very relevant to our work, which the reviewer pointed out. The *Nature 2017* article reports a self-oscillating film that, coupled to a very unique “device” design, translated the self-oscillation into an autonomous walking device. In the revised manuscript, we thoroughly discuss the challenges of such mechanical integration, and give prospects for light-driven self-oscillating robots for future research. The *Nat. Commun. 2016* paper reports an interesting chaotic oscillating phenomenon, without prototype demonstration. Alike that paper, we believe the present paper’s most important asset to lie in novel finding, the freestyle oscillator, which itself will attract the broad interest in the research community even without crystal-clear application prospects.

1. In Figure 2d (contraction mode), light is incident from the upper perpendicular to the upper surface, so the lower surface is not illuminated. Does this difference in light irradiation between the upper and lower surfaces lead to some bending deformations?

In the contraction-expansion mode oscillation, the difference in light excitation does not lead to bending deformation because of the following reasons. First, bending is a dimension-related effect. For efficient bending, the film should have a large length-to-thickness ratio. This is why the samples used in bending and twisting modes are both film-like, with length-to-thickness ratio around 100. For contraction-expansion mode, we have chosen a microscopic LCN fiber with 100 μm diameter. The deformation length is in the same order as the thickness (pay attention that the focused laser only actuates a small portion at the tip, not the whole fiber). Given these dimensions and irradiation conditions, the bending of the fiber is suppressed. Secondly, the actuation is due to photothermal effect, i.e., not driven by photo-isomerization that contributes to the actuation in bending/twisting modes. The photo-isomerization can lead to a large gradient in *cis*-isomer population across the sample thickness, due to limited light penetration depth, which enhances the bending actuation. However, upon photothermal excitation, the temperature between the upper and lower surfaces will be identical due to fast heat conduction occurring at a microscopic scale, hence no significant gradients are formed.

We agree with Reviewer that the actuation differences between the bending and contracting actuators are very important, and have thus added the following details into the Methods section:

Bending and twisting actuators are film-like, with length-to-thickness ratio about 100. For contraction-expansion mode, a microscopic fiber with 100 μm diameter was used, with length-to-thickness ratio around 1. Because of the low length-to-thickness ratio, the bending of the fiber was suppressed. / The temperature difference between illuminated and non-illuminated surfaces is negligible due to fast thermal transport occurring at microscopic scale.

2. In page 13, line 247, “The strip (Fig. 3b, i) bends when illuminated with a laser spot (100 mW) at the centre of the cantilever (Fig. 3b, ii).” This way for realizing bending mode is different from the description shown in figure 2 “Bending is obtained by light propagation along the cantilever axis, in which case the laser alternately activates both surfaces during the cyclic motion.” Please give the reason.

The film used for the bending mode (Fig. 2, a, b, c) yields photochemical actuation, and the laser beam can bend the strip to both sides. Hence we set the laser beam to propagate along the cantilever axis, to activate the bending towards both directions. In the freestyle mode, the actuation is based on photothermal effect and the bending occurs to the same side no matter which surface is illuminated. It is indeed very important to clearly explain the photomechanical differences between the samples used. Hence we have further elaborated this aspect in the Methods Section of the revised manuscript as follows:

Photoactuation in brief. An unfocused, continuous-wave 488 nm laser beam was propagating along the axis of the LCN cantilever in case of bending-mode oscillation, and perpendicular to the axis for the twisting mode. Upon laser excitation, the azobenzene moieties in the LCN undergo repeated trans-cis-trans isomerization. Due to limited light penetration depth, isomerization gradient across the film thickness is formed which, together with photothermal heating, gives rise to cantilever deformation towards the light source. As a result, the laser beam illuminates alternately both cantilever surfaces, yielding periodic bending/twisting. Further details on the optical configuration are given in Supplementary Information. For contraction-expansion -mode oscillation we used a purely photothermally driven actuator, where light was absorbed at a tip of an LCN fiber, the energy was transferred into heat, and conducted along the cantilever to the rest of the actuator body. / The LCN used for freestyle oscillation was also a photothermal actuator. Because of its large deformability, multiple degrees of freedom of deformation can be activated by impinging the light beam at different sample positions and/or by changing the excitation power.

3. In page 12 line 250, the authors mentioned that “By increasing the light intensity from 100 to 140 mW, a larger extent of deformation can be triggered, and bending and contraction occur simultaneously (Fig. 3b, iv).” Without changing the position of illumination, the oscillation mode can be change only be increasing the light intensity ? It’s different from the previous description of the contraction mode, why?

First of all, we would like to point out that the freestyle actuator (Fig. 3) and contracting actuator (Fig. 2) are made of different LCN compounds, and more importantly, their deformation degrees of freedom are very different. Secondly, in Fig. 3b, there is no oscillation. The figure aims to show that by changing the intensity and position of the laser beam, one can induce bending, twisting and contracting deformations simultaneously within one single LCN strip. Conversely, the samples in Fig. 2 only exhibit one specific deformation mode (bending, twisting or contraction). In contraction-expansion oscillators, the deformation mode is fixed: the actuator lacks the degrees of freedom required for bending or twisting. Supplementary information and newly included experiments to address the comments of Reviewer 1 show that both frequency and amplitude can be changed by varying laser intensity or position. In the freestyle oscillator, the deformation mode, frequency and amplitude all can be changed, as proved by projected trajectories (change of deformation modes, Fig. 4b). Change of modes occurs not only by varying the laser intensity, but also spontaneously without changing the illumination conditions (Fig. 4c).

4. The authors explain that the final stop of the self-oscillation is resulted from the irreversibility of materials response. The more specific explanations should be provided. What about the other oscillation such as bending, contraction and twisting? Do they finally also reach the stable position under the constant light illumination?

Thanks for an important comment. Irreversibility is one of the fundamental features of all cyclically actuating “smart” materials. It arises from the fact that upon cyclic shape-changes, the polymer chains are being continuously stretched and coiled up, a process which eventually leads to irreversibility in their deformation capability. In case of photoactuation, bleaching of the photoactive dyes is another factor contributing to the irreversibility. We now explain this in more detail, as shown below:

The reason for such photomechanical irreversibility may be attributed to (i) kinetic deformation driven by elastic entropy, where cyclic stretching-coiling gives rise to friction between the polymeric chains, and (ii) photobleaching of the dyes. / In all cases, the excitation beam position needs to be finely tuned in order to reach the stable self-oscillating movement. Once self-oscillation is reached, it lasts typically > 30 min for bending and twisting modes, > 50 min for contraction-expansion mode and > 30s for the freestyle mode. More details on photomechanics and the optical configurations used for different oscillation modes are given in the Supplementary Information.

5. In figure 2, different oscillation modes were realized by using different LCN materials with different dimensions. During this self-oscillation process, the laser beam is illuminated on the different position of different materials, and the spot size of the laser beam are all different. In contrast, in figure 3, the different oscillation modes are achieved by only changing the position of the laser spot on one same sample. The more detailed analysis and explanation must be provided.

Thanks for the constructive comments. LCN materials are well known for their diversity in terms of the wealth of deformations, determined by their composition and alignment, they can produce. Hence, to realize basic deformation modes (bending, twisting and contraction-expansion) separately, we have to fabricate LCNs with different dimensions and alignment, and carefully choose the irradiation conditions (Fig. 2). The dependence of the deformation on molecular alignment and the optical configuration are well studied in the literature. For clarification, we have provided detailed explanation about photomechanics between different actuators in Methods and in Supplementary Information. For the basic deformations given in Fig. 2, we design the actuators to only exhibit one specific mode of deformation, while the other two basic deformation modes are suppressed due to LCN dimensions/alignment. Note that the laser spot size used was identical for bending, twisting and freestyle oscillation, while for inducing contraction, we used a focused laser beam due to smaller dimensions of the LCN. To obtain self-oscillation with combined deformation modes, a softer (main-chain) LCN with larger deformability was used (Fig. 3). In this situation, all the deformation modes are active within one LCN strip, simply by inducing a laser beam excitation at different positions.

6. Does the evolution of the self-oscillation can be stably modulated from a mode to another certain mode?

Until now, we have not been able to control the self-oscillation modes, neither predict or tune the evolution between certain modes. We have developed a tracking system to record the evolution as shown in Fig. 4. The lack of control is due to the multiple degrees of freedom in deformation, and the complexity

of light-matter interactions in soft materials, resulting in different time delay in materials response at different stages of the oscillation cycle. We have added the following discussion to explain this:

The prediction and control of the evolution between different modes during the freestyle oscillation is difficult to attain, due to the multiple degrees of freedom in deformation and complexity of light-matter interactions in responsive soft materials, resulting in different time delay in materials response at different stages of the oscillation cycle. Despite the lack of control, the intriguing free-style self-oscillation phenomenon provides insight into the process which, we believe, can be generalized to different kinds of responsive materials with arbitrary geometry, deformability, and stimuli-response, as long as a feedback mechanism is established for sustaining the periodic motion.

7. The application examples of the self-oscillators with different oscillation modes should be given to demonstrate their practical application prospect.

This manuscript, for the first time, reports a freely deforming, irregular strip (bending, twisting and contacting simultaneously) that can self-oscillate under a constant light beam. The results give fundamental insights into self-oscillation in active, responsive materials. We also point out that one of the references highlighted by the Reviewer (*Nat. Commun.* **2016**, 7, 11975) also doesn't contain any practical prototype, however the phenomenon itself has attracted broad readership (highly cited).

However, in the revised manuscript, we demonstrate a novel device prototype for optical applications, based on a self-oscillating strip. We realize a light-powered beam modulator that can modulate one-beam transmission at its self-oscillation frequency (33Hz, transmission from 5% to > 90 %), and even modulate two beams simultaneously, selecting them to be either in-phase or 180° out-of-phase. More details on the revision are given below.

Beyond the above-mentioned robotic applications that aim to transfer input light energy into mechanical work output, there is a frontier of optical applications to be investigated. A self-oscillator is essentially an optical chopper. However, compared to the conventional electronic motorized chopping system, it has many advantages such as compact size, light weight, and, being remotely powered, possibility towards integration into micro-devices. An oscillating strip can modulate a signal beam (635 nm) periodically in its transmission, as shown in Fig. 5a: during the down-bending stage the beam is transmitted while the up-bending stage blocks the beam. In other words, the signal beam transmission is controlled and modulated by the self-oscillator (Fig. 5b). Furthermore, one single strip can even control two signal beams simultaneously. As shown in Fig. 5c, the two signal beams can be in phase, when propagating along the same side of the strip, or 180° out-of-phase, when propagating from the opposite sides of the strip (Fig. 5d).

Fig. 5. Self-oscillator-based laser beam modulator. (a) A self-oscillating strip (bending mode, driven by a continuous 488 nm, 190 mW) can modulate the transmission of a signal beam (635 nm). The signal beam is focused on the upper part of the oscillator as shown by the schematic drawing, while the transmitted beam profiles are shown in the photos. Scale bar: 1 cm. (b) Transmission measurement of the beam being modulated by the self-oscillator. (c) When two signal beams are propagating along the same side of the oscillator, their transmission is modulated in phase. (d) When two signal beams are propagating along the opposite sides of the oscillator, their transmission is modulated 180° out-of-phase. Blue curves in (c) and (d) show the scattered 488 nm light intensity for phase reference. Strip size: $1 \times 10 \times 0.05 \text{ mm}^3$.

REVIEWERS' COMMENTS:

Reviewer #1 (Remarks to the Author):

Despite my remarks in my earlier review of the manuscript I was already quite positive towards publications of the manuscript in Nature Communications. In their rebuttal the authors addressed most of my concerns. Only my remark that some of figures show samples that in my opinion are somewhat below the standards that are normally achieved with similar materials are confuted by the authors by stating that they demonstrated the information to prove their proposed principles and mechanisms. This is a matter of opinion and taste that I leave for the editor to take a decision on.

The authors extensively added information to the manuscript in response to the three reviewers. Consequently the manuscript expanded and might be shrunk again to meet the length of a communication. But also this is for the editor to decide.

For the remaining I am very positive towards publication.

Reviewer #2 (Remarks to the Author):

I am happy with the revisions and responses made by the authors, both to my points and to the reasonable queries of the other referees too.

The paper should be published.

Reviewer #3 (Remarks to the Author):

The paper is well revised and recommended to be accepted.

Answers to the Reviewers' comments

Reviewer #1

Despite my remarks in my earlier review of the manuscript I was already quite positive towards publications of the manuscript in Nature Communications. In their rebuttal the authors addressed most of my concerns. Only my remark that some of figures show samples that in my opinion are somewhat below the standards that are normally achieved with similar materials are confuted by the authors by stating that they demonstrated the information to prove their proposed principles and mechanisms. This is a matter of opinion and taste that I leave for the editor to take a decision on.

The authors extensively added information to the manuscript in response to the three reviewers. Consequently the manuscript expanded and might be shrunk again to meet the length of a communication. But also this is for the editor to decide.

For the remaining I am very positive towards publication.

We thank Reviewer for his/her comments and for the positive viewpoint towards publication.

About the image quality: All the photos used in this manuscript are taken with the best resolution and quality that can be achieved during our experiments and we don't see a way of improving them further. We would like to kindly note that the small-sized photoactuator strip is rapidly moving/oscillating in three dimensions, and simultaneously exposed to light with relatively high intensity. These inevitably reduce the picture quality compared to those achieved in conventional, or more static, LCN actuators. Second, we would like to emphasize that all the figures, including the experimental photographs and schematic drawings, in our opinion, clearly bring out the focus of this work, which is the principle of self-oscillation with different degrees of freedom of deformation and different oscillation modes.

About the length of the manuscript: The word count for the main text, including all figure captions, is 4900. The count for introduction, including abstract, is 950. The manuscript contains 5 figures. All these should be in line with the standards of *Nature Communications* articles. Therefore, no adjustment to the length of the manuscript is made in the revised version of the manuscript.